# Facile Use of ZnO Nanopowders to Protect Old Manual Paper Documents

**DOI:** 10.3390/ma13235452

**Published:** 2020-11-30

**Authors:** Ludmila Motelica, Aurelian Popescu, Anca-Gabriela Răzvan, Ovidiu Oprea, Roxana-Doina Truşcă, Bogdan-Stefan Vasile, Florina Dumitru, Alina-Maria Holban

**Affiliations:** 1Faculty of Applied Chemistry and Material Science, University POLITEHNICA of Bucharest, 060042 Bucharest, Romania; motelica_ludmila@yahoo.com (L.M.); anca_razvan@yahoo.com (A.-G.R.); truscaroxana@yahoo.com (R.-D.T.); bogdan.vasile@upb.ro (B.-S.V.); florina.dumitru@upb.ro (F.D.); 2Department of Publications Restoration and Preservation, “Carol I” Central University Library, 010292 Bucharest, Romania; aurelian.popescu@bcub.ro; 3Microbiology & Immunology Department, Faculty of Biology, University of Bucharest, 077206 Bucharest, Romania; alina.m.holban@bio.unibuc.ro

**Keywords:** ZnO, paper, antimicrobial activity, long time protection, photocatalytic activity

## Abstract

One of the main problems faced by libraries, archives and collectors is the mold degradation of the paper-based documents, books, artworks etc. Microfungi (molds) emerge in regular storage conditions of such items (humidity, usually over 50%, and temperatures under 21 °C). If the removal of the visible mycelium is relatively easy, there is always the problem of the subsequent appearance of mold as the spores remain trapped in the cellulosic, fibrillary texture, which acts as a net. Moreover, due to improper hand hygiene bacteria contamination, old books could represent a source of biohazard, being colonized with human pathogens. An easy and accessible method of decontamination, which could offer long term protection is therefore needed. Here, we present a facile use of the ZnO nanopowders as antimicrobial agents, suitable for cellulose-based products, conferring an extended antibacterial and anti-microfungal effect. The proposed method does not adversely impact on the quality of the cellulose documents and could be efficiently used for biodegradation protection.

## 1. Introduction

The life expectancy of collections is significantly affected by environmental storage conditions. Improper storage conditions for books or various documents on cellulosic or parchment paper can cause bacteria, fungi, or mold to grow on their surface [1]. The growth and multiplication of these pathogens leads to the production of digestive enzymes that alter and weaken the structure of the organic support, leading to the appearance of spots or even more severe degradation [2]. Spores eliminated by these pathogenic species also pose an important risk to human health. At present there are several possibilities for treating books in which sheets of paper or parchment have been degraded by bacteria, yeast and filamentous microfungi (molds). The support medium of the document can be brushed with a soft brush, vacuumed with HEPA (high-efficiency particulate air) system, or placed in a dry freezer. None of these methods ensures a deep removal of spores, nor does it provide a subsequent resistance of the document to pathogens. Another method used involves treatment with ethylene oxide or thymol, these substances being effective in removing the mold, but unfortunately, they are also carcinogenic [3].

At the individual level, some facile methods are applied, such as drying with a hairdryer, packing the document in a bag with rice or flour. These methods could only ensure the elimination of excess moisture absorbed by the cellulose support. Sodium bicarbonate packaging can be used to remove the odor generated by the present molds. None of these methods solves the problem of the existence of mold or its spores. If bacteria are present on a document, the owners usually do not even notice it as the manifestations are not as visible as in the case of molds [4]. Numerous bacterial species, especially gut-derived, present cellulose–degrading ability, similar to microfungi. Such microorganisms have cellulolytic potential and present metabolic degrading enzymes for cellulose containing foods, which reach in human digestive system [5]. Human opportunistic pathogens could colonize old books and cellulose based artworks by air-borne and improper hand hygiene contamination.

Two problems can be identified at this time, one economic and one technical. The methods currently used to process various records on cellulosic support or parchment (books, documents, official individual papers or from archives etc.) involve high cost equipment and require specialized staff. None of the current cleaning methods ensures a long-term protection of the cellulose based valuable products, requiring periodic interventions on the respective documents. In order to have a lasting antimicrobial effect, agents that provide antifungal and antibacterial activity must remain in contact with the cellulose product (i.e., document), without damaging it, and without representing a health hazard for the people who come in contact with the document.

The ZnO is one of the most studied nanomaterial, with applications in various domains, based on couple of its properties. Due to its high absorbance in the UV region, ZnO is commonly used as sunscreen [6], due to its intrinsic antimicrobial activity is used in the food industry and in medicine (also as drug delivery system) [7], and due to its transparent, semiconductor nature ZnO is used in optoelectronic devices [8], and finally as a pigment, ZnO is used in various paints or anticorrosion coatings [9]. The antibacterial activity of ZnO has been known for some time, but the antimycotic activity is seldom investigated [10,11,12,13,14]. The exact antimicrobial mechanisms is not known, but there are at least two separate pathways with respect to the light presence. The first mechanism implies photocatalytic activity, and perhaps reactive oxygen species (ROS) production, which is responsible for oxidative stress that is damaging the bacterial membrane. The second mechanism does not require light and is based on nanoparticle internalization and mechanical damage (like puncture and rupture) to the microorganism cellular wall [15,16,17].

Generally, ZnO nanoparticles (NPs) or bulk ZnO particulates, when used as antibacterial agents, are considered bio-safe within certain limits, which depend on the availability or concentration of zinc ions, but raise safety issues at higher concentrations [18], especially in nano form when ingested [19]. ZnO NPs have widespread use in cosmetics [20], various paints and coatings [9], active food packaging [21] and numerous biomedical applications [22], and thus, hazards from exposure via inhalation [23] and dermal exposure to them are of great concern and numerous studies have been carried out assessing their cytotoxicity when ingested or applied onto human skin [24,25,26]. The results are still controversial because of the information scarcity on human exposure assessment in realistic use [27], since the majority of studies were focused on the first two steps of risk assessment process; the intrinsic toxicity and dose-response relationship of ZnO NPs [28,29]. Although, ZnO NPs exhibited in vitro experiments oxidative stress-induced apoptosis in human lung epithelial cells (L-132) [23] or neurotoxic potential on neuronal cells [24], the cytotoxicity of ZnO NPs in these studies is evaluated at concentrations that are likely higher than would be expected for short-term exposure (topical or inhalation) [30]. There is widely accepted that ZnO NPs do not penetrate healthy human skin, even they do affect skin cells in vitro [25,27]. Furthermore, all in vivo and most in vitro genotoxicity tests were negative for ZnO NPs [28,31,32].

Taking into account all these considerations, the current paper reports, the successful use of ZnO nanoparticles as antimicrobial agent, being efficient against common human opportunistic pathogens, including microfungi, which colonized old documents. When handling such old books, the users should anyway wear gloves, thus, limiting contact both, with pathogens and ZnO NPs, and in the absence of gloves the exposure time is relatively short. The ZnO nanoparticles were characterized by transmission electron microscopy (TEM), X-ray diffraction (XRD), ultraviolet-visible (UV-Vis) and photoluminescence (PL) spectroscopy. The photocatalytic and antibacterial activities were also determined. The paper samples treated with ZnO nanoparticles were investigated by UV-Vis and PL spectroscopy, Fourier transform infrared (FTIR) microscopy.

## 2. Materials and Methods 

### 2.1. Materials

Zinc acetate dihydrate with 99.9% purity was obtained from Merck (Merck Group, Darmstadt, Germany). The absolute ethanol was used as received from Sigma (Redox Lab Supplies Com SRL, Bucharest, Romania), without further purification. ZnO synthesis was done as described in [33]. Briefly, 2.1950 g zinc acetate dihydrate were solved in 50 mL absolute ethanol and heated under magnetic stirring at the boiling point. After 10 h, the precipitate was washed and centrifugated three times, and the resulting powder was dried at 105 °C.

Two handmade paper sheets, from XVIII-XIX century, were received from “Carol I” Central University Library, Bucharest, and were noted as sample A (printed paper) and sample B (blanc paper). Each sheet was cut in strips to permit multiple analysis. Some of the paper strips were treated with ZnO by sprinkling minute quantities of nanopowder on the paper surface, followed by gentle brushing, with linear moves, by a soft brush with 50 mm long bristles. The excess powder was discarded by gentle tapping.

### 2.2. Characterization of ZnO Nanoparticles and Paper Samples

The transmission electron images for ZnO nanoparticles were obtained on dried, finely powdered samples using a Tecnai G2F30 S-TWIN high-resolution transmission electron microscope from FEI (FEI Company, Eindhoven, The Netherlands), operated at an acceleration voltage of 300 kV obtained from a Schottky field emitter with a transmission electron microscope (TEM) point resolution of 2 Å and line resolution of 1.02 Å.

Scanning electron micrographs for determination of the paper surface morphology and microstructure were obtained by using a QUANTA INSPECT F50, FEI Company, Eindhoven, The Netherlands scanning electron microscope equipped with field emission gun (FEG) with 1.2 nm resolution and an energy dispersive X-ray spectrometer (EDS, Thermo Fisher—formerly FEI, Eindhoven, The Netherlands) with an MnK resolution of 133 eV

In order to investigate the crystalline phases of the obtained nanopowder, the sample was analyzed using the PANalytical Empyrean equipment (from Malvern PANalytical, Bruno, The Nederland) using a Bragg–Brentano geometry, equipped with a Cu anode (λ_CuKα_ = 1.54184 Å) X-ray tube and hybrid monochromator (Ge220). The X-ray diffractogram (XRD) was acquired in the 2θ range 10–75°. An acquisition step of 0.02° and an acquisition time of 100 s per step were employed.

The photoluminescence spectrum (PL) was measured with a Perkin Elmer P55 (Perkin Elmer, Waltham, MA, USA) spectrometer using a Xe lamp as a UV light source at ambient temperature, in the range 350–600 nm, with the sample dried and finely powdered. The measurement was made with a scan speed of 200 nm min^−1^, slit of 10 nm, and cut-off filter of 1%. An excitation wavelength of 320 nm was used.

Diffuse reflectance spectra measurements were made with a JASCO (Easton, PA, USA) V560 spectrophotometer with a solid sample accessory, in the domain 200–800 nm, with a speed of 200 nm min^−1^.

The photocatalytic activity was determined against methylene blue (MB) 3.125 × 10^−5^ M (10 mg/L) solution, by irradiation with a LOHUIS^®^ (Ulmi, Romania), commercially available, fluorescent lamp of 160 W/2900 lm (lumen), with color temperature of 3200 K and color rendering index >60, placed at 20 cm distance. The sample of 0.0250 g powder was inserted in 10 mL solution of MB and left under stirring for 30 min in dark to reach adsorption-desorption equilibrium. After irradiation, at defined time intervals, a sample of 3 mL was placed in a quartz 10 mm cuvette, and its UV-Vis spectra was recorded.

In order to obtain information about the spatial distribution of the ZnO nanoparticles on a paper surface, FTIR two-dimensional (2D) maps were recorded with a FTIR microscope Nicolet iS50R (Nicolet, Waltham, MA, USA), with DTGS detector, in the wavenumber range 4000–400 cm^−1^.

### 2.3. Antimicrobial Evaluation

#### 2.3.1. Antibacterial Activity

Qualitative antibacterial activity of ZnO nanoparticles was tested against Gram-negative *Salmonella typhimurium* (ATCC 14028), *Pseudomonas aeruginosa* (ATCC 27853), *Escherichia coli* (ATCC 25922) and Gram-positive *Staphylococcus aureus* (ATCC 25923), *Bacillus subtilis* (ATCC 6633) and *Enterococcus faecalis* (ATCC 44479) bacteria, by drop inoculation on nutrient agar.

Briefly, the tested bacterial strains were swab inoculated on nutrient agar medium in Petri dishes and drops with a volume of 5 μL of a 10 mg/mL ZnO suspension were added. The Petri dishes were incubated 24 h at 37 °C and after that inhibition diameters were assessed. All experiments were designed and performed in triplicates.

#### 2.3.2. Microfungal (Mold) Removal

In order to assess the ability of the obtained ZnO nanoparticles to remove molds colonizing cellulose-based products, samples of old paper from books presenting microfungal colonization and degradation were cut into pieces of 1 cm/1 cm with sterile instruments to avoid external contamination. Then, some of the obtained old paper samples were treated with the ZnO nanoparticles, and some were used as control. The obtained specimens were used to inoculate PDA (potato dextrose agar) plates by a three-point inoculation method. Inoculated PDA was incubated in the dark for up to 7 days, at 20–22 °C.

## 3. Results and Discussion

ZnO nanoparticles have been prepared in a non-basic route: a controlled solvolysis of Zn(CH_3_COO)_2_·2H_2_O in mild conditions (ethanol boiling temperature, no alkaline solution, water or other chemicals added) and therefore, the premises for their use in paper conservation are fully met. By weighting rectangular samples of paper (with various known dimensions) before, and after, applying ZnO nanopowder, we calculated that the retention of ZnO nanoparticles into the cellulosic support is on average 0.23 ± 0.01 mg/cm^2^ on each side.

### 3.1. Transmission Electron Microscopy (TEM)

The TEM bright-field image obtained on ZnO nanoparticles synthesized by the solvothermal method is presented in Figure 1.

The Figure 1a–c images reveal that the powder is composed of polyhedral-shaped nanoparticles, with an average grain size of approximately 20 nm, as can be observed from the histogram presented in Figure 2b.

The histogram was obtained by measuring about 500 nanoparticles, and only when they had clearly defined edges. The extreme values were discarded for statistical reason. Nevertheless, some nanoparticles have higher diameter of ~30 nm. The nanopowder also has the tendency to form some soft agglomerates. From the selected area diffraction (SAED) pattern obtained on ZnO nanopowder, we can state that the only phase identified is the polycrystalline hexagonal form of ZnO [JCPDS card no. 80-0075]. The high-resolution transmission electron microscopy (HRTEM) image obtained on nanocrystalline ZnO is shown in Figure 1d. The image shows clear lattice fringes of polycrystalline nanopowder of d = 2.59 Å corresponding to the (0 0 2) crystallographic planes of ZnO. Also, the regular succession of the atomic planes indicates that the nanocrystallites are structurally uniform and crystalline with no amorphous phase present.

Energy-dispersive X-ray spectroscopy (EDX) was used to determine the composition of the sample and the EDX pattern (Figure 2a) confirms that the isolated ZnO nanopowder is pure and contains only zinc and oxygen.

### 3.2. X-ray Powder Diffraction (XRD)

Figure 3 presents the XRD pattern for ZnO nanopowder prepared by the solvothermal method.

All the peaks correspond to the hexagonal, wurtzite, ZnO structure [JCPDS card no. 80-0075]. The XRD pattern indicates that the sample is pure and crystalline, with no second phase being identified.

The obtained XRD data was fitted by Rietveld refinement in order to determinate the crystallite size and microstrain. The obtained values are crystallite size D = 26.75 ± 3.03 nm, and microstrain ε = 0.164 ± 0.050%. The crystallite average size is slightly larger than the TEM histogram value, due to statistical discard of outlier particles, but results suggest that each nanoparticle is composed of a single crystallite.

The lattice constant calculated for the ZnO nanoparticles are *a* = *b* = 3.2494 Å and *c* = 5.2051 Å, with a ratio *c/a* = 1.6018.

### 3.3. UV-Vis and PL Spectrometry

The electronic spectrum recorded for the ZnO nanopowder is presented in Figure 4a.

While, ZnO presents a very low absorption in visible region, the characteristic high intensity absorption band from UV regions present a maximum at 358 nm. The fundamental absorption is related to the optical transition of electrons from valence band to conduction band and can be used to calculate the optical band gap of the ZnO nanopowder.

The diffuse reflectance of the sample, R, can be related to the Kubelka-Munk function F(R) by relation, Equation (1):F(R) = (1 − R)^2^/2R.(1)

The direct band-gap energy value for the ZnO nanopowder was calculated from a plot of the square of the modified Kubelka-Munk function vs. energy (eV), the inset of Figure 4a. The value was determined to be 3.19 eV, by extrapolation to [F(R)∙*hν*]^2^ = 0 [34]. The calculated band-gap energy is close to the theoretical value of 3.37 eV, but is smaller due to multiple defects that are inducing new electronic levels inside the ZnO band gap. Similar lower values are reported in literature by [35,36,37,38].

The recorded photoluminescence spectra of ZnO is presented in Figure 4b. Usually the ZnO nanopowders present two emission zones, one in UV and one in visible region. The UV emission corresponds to the near band edge emission (NBE) and is generated by the free exciton recombination. The sizeable binding exciton energy of the ZnO (60 meV) is responsible for the room temperature photoluminescence of ZnO [39]. The visible emission is usually referred as deep level emission (DLE) and is related to the existence of defect generated electronic levels inside the band gap (Figure 5). Calculating the ratio between the integrated area of NBE and DLE bands can be used to estimate the defect levels contribution. For high crystalline ZnO nanopowders the NBE is more intense than the DLE [40].

The ZnO crystal defects can be zinc vacancies (V_Zn_), zinc interstitials (Zn_i_), oxygen interstitials (O_i_), oxygen anti-sites (O_Zn_) or oxygen vacancies (V_O_) [41]. From these, oxygen vacancies are deep rather than shallow defects and can be involved, as recombination centers for green emission in ZnO, with lower energy for V_O_ site (placed upper in the band gap) and higher energy for V_O2+_ site. The surface excess oxygen or -OH moieties are acting as donor centers for the V_O_ sites, and a weaker green emission indicates a lower surface defect concentration [42]. Zinc interstitials and zinc anti-sites are also shallow donors, but the last type requires a very high formation energy [43]. Zinc vacancies are deep acceptors and usually are related to the green and blue emissions. Oxygen interstitials are also deep acceptors with high formation energy (usually at octahedral interstitial site).

The obtained ZnO nanopowder has a strong NBE emission at 3.25 eV (381 nm), with peaks from DLE at 2.67 eV (465 nm), 2.57 eV (482 nm) and green emission at 2.43 eV (509 nm). This confirms the high crystallinity of the sample, as indicated by TEM and XRD. The V_Zn_ are responsible for the emission in the violet region, transitions from Zn^+^_i_ to valence band (VB) generate the 2.67 eV (465 nm) band, while transitions from Zn^+^_i_ to V_Zn_ are responsible for the 2.57 eV (482 nm) emission. The green emission from 2.43 eV (509 nm) and the further is generated mainly by V_O_ and V_O+_ defects [44].

### 3.4. Photocatalytic Activity

The photocatalytic activity of ZnO nanopowder is well-known [45,46,47], along with that of TiO_2_ [48,49], and the literature reports many investigations, due to possible links with antimicrobial activity and photoluminescence properties. We chose to investigate the photocatalytic activity against methylene blue (MB), as this is a phenothiazine dye and has the tendency to dimerize. Therefore, the photocatalytic activity is investigated against the monomer with absorption maximum at 664 nm, but also vs the dimer with absorption maximum at 614 nm.

The decrease of both absorption maxima, Figure 6, indicate that ZnO nanoparticles are capable to photodegrade both organic compounds.

For diluted solutions the photodegradation reactions exhibit apparent first-order kinetics, ln (C0/C) = *k*_app_∙t, where C0 is the initial concentration of MB, C is the MB concentration at time t (hours) and *k*_app_ is the rate constant of apparent first order. Similar first-order kinetics fitting are reported for MB degradation by ZnO in [50,51,52]. The calculated value for *k*_app_ from the slope of the inset graphic is 0.2014 ± 0.0192 h^−1^, with R-square value of 0.940. The values obtained for the rate constant are dependent on multiple factors, among which MB initial concentration and ZnO quantity. Comparable rate constant value is reported for concentration of 15 mg/L MB and 75 mg photocatalyst [51]. The same paper reports that decreasing the concentration to 10 mg/L, while keeping the ZnO quantity at 75 mg, more than doubles the rate constant, while further decreasing the concentration to 5 ppm produce no significant changes. This suggest that the excessive coverage of catalyst surface with MB molecules can act as a light-barrier, and prevents the immediate degradation of adsorbed molecules [50]. For solution with 10 mg/L MB put in contact with 50 mg oxygen defect-rich ZnO under visible light irradiation, the reported *k* value is 5.198 × 10^−3^ min^−1^ (0.3118 h^−1^), while for ZnO prepared by basic precipitation the *k* value is 4.0337 × 10^−4^ min^−1^ (0.0242 h^−1^), which suggest that the oxygen defects plays also an important role in the photodegradation of the dye [53].

As we mentioned the antibacterial mechanism of ZnO nanoparticles is still unclear, but it exhibit different activities in the presence of light and under dark [54]. In presence of light, reactive oxygen species (ROS) are generated and photocatalytic activity might be involved into the promotion of the antimicrobial activity. ROS are responsible of photocatalytic activity, but also for the oxidative stress which is damaging the bacterial membrane [55].

### 3.5. Antimicrobial Activity

Agar diffusion tests were performed as a qualitative rapid method to observe the ZnO nanoparticles antibacterial activity.

Antibacterial activity of ZnO nanoparticles was tested against opportunistic bacteria species, such as Gram-negative *Salmonella typhimurium*, *Pseudomonas aeruginosa*, *Escherichia coli* and Gram-positive *Staphylococcus aureus*, *Bacillus subtilis* and *Enterococcus faecalis* strains. The qualitative results presented in Figure 7, indicate a good antibacterial activity, the least susceptible being *P**. aeruginosa* which is well known also for its low antibiotic susceptibility.

In numerous studies on the ZnO nanoparticles, antibacterial activity is correlated with different particle morphologies [56]. The shape-dependent activity is most probably related to the percent of active facets of the nanoparticles. As we obtained the polyhedral nanoparticles with size ~20 nm the observed antibacterial activity is good, the ZnO nanoparticles being effective against a broad spectrum of bacterial strains.

### 3.6. Scanning Electron Microscopy (SEM) of Old Documents Paper

The SEM micrographs for two paper samples (XVIII-XIX century) received from “Carol I” Central University Library are presented in Figure 8.

The images revealed the high infestation degree of these samples with various microorganisms (most of them belonging to *Aspergillus* sp.) [57]. In libraries, fungal growth known as mold or mildew affects paper, leather and textiles. Fungi consume cellulose and thrive on nutrients from gelatin size, collagen from leather, glues from binding etc. They can weaken and stain the paper, causing foxing, discoloration and in the end the destruction of the document support. Some bacterial strains also decompose the cellulose in paper and binding textiles [58].

The EDS elemental mapping images of the ZnO treated paper clearly indicate the uniform distribution of the ZnO nanoparticles on the surface (Figure 9).

The SEM images indicate that the ZnO nanoparticles formed small, spherical agglomerates that are presented among the cellulose fibers.

### 3.7. FTIR Microscopy of Old Documents Paper

Samples of ~10 mm × 10 mm were cut from the paper strips after ZnO application. The spatial distribution of ZnO nanoparticles on the paper surface after treatment was investigated by FTIR microscopy. The FTIR maps recorded at 430 and 1643 cm^−1^ for the two paper samples are presented in Figure 10. As Figure 10a indicate, the ink is clearly visible after applying the ZnO treatment.

The wavenumbers were chosen as they are characteristics for stretching vibration of Zn-O bond (430 cm^−1^) and for C=O stretching in amide I band (1643 cm^−1^) [59,60]. The old paper manufacturing process required sizing of the paper surface with gelatin, hence the amide group presence [61]. The distribution of ZnO nanoparticles on the surface of the cellulosic material is quite homogenous, with occasional small agglomerated clusters/accumulations. There are some noticeable differences between the maps recorded at 430 and 1643 cm^−1^ for sample A_ZnO, which means that even if ZnO nanoparticles are homogeneously distributed on the sample surface (Figure 10c), the ink from the letter is still visible (as can be seen in Figure 10a,b). This indicate that after treating the paper, letters are clearly visible, while some ZnO nanoparticles are being obviously retained on the surface.

The FTIR maps for sample B_ZnO, Figure 10b,c, suggest also a homogeneously distribution of ZnO on the paper surface, with small zones in which are presented some agglomeration. The ZnO NPs most probably interact with cellulose fibrils via -OH moieties as indicated in [62,63]. The FTIR maps suggest that ZnO application method is successfully distributing the nanoparticles between cellulosic fibers, which also acts like a net, the quantities applied not interfering with ink visibility.

### 3.8. UV-Vis and PL Ppectrometry on Old Documents Paper

Strips from the samples A and B are presented in Figure 11, before, and after, treatment with ZnO nanoparticles. In order to better visualize the differences between the treated and untreated surface, the treatment was performed initially only in the selected areas. As can be observed, the paper became slightly whiter, and the foxing which was present on the sample A disappeared.

Both samples (A and B) were light yellow due to their old age and to the manufacturing process. The UV-Vis and PL spectra were recorded on areas indicated in Figure 11. The paper samples had quite different spectra in UV region, Figure 12, with absorption maxima at 220 and 322 nm, indicating both different compositions and manufacturing processes.

For the samples treated with ZnO nanoparticles (A_ZnO and B_ZnO), both UV-Vis spectra are similar and present one broad, very intense absorption band in the UV region, with a maximum at 360 nm. The shape of the UV-Vis spectra is attributed to the presence of ZnO nanoparticles on the paper’s surface. Moreover, this indicates that ZnO nanoparticles can also act as a UV-barrier and effectively protect the paper from harmful UV radiation. The intensity of the absorption band is higher than the initial ZnO powder, indicating a good dispersion between cellulosic fibers and a good scattering of incoming radiation. The existence of some interactions between nanoparticles and fibrils, leading to the increase of absorbance, is also highlighted by the different intensities of A_ZnO and B_ZnO absorbances.

A whitening effect of treatment with ZnO nanoparticles of the samples is also observed in the visible region of spectra.

The photoluminescence spectra for treated and untreated samples are presented in Figure 13.

The untreated manual paper had a high, characteristic blue fluorescence due to the presence of sizing gelatin [64,65] and from cellulose fibrils [66,67]. Both samples present a large, intense, violet-blue emission band, with multiple peaks and a tail towards longer wavelengths.

As the fluorescence is generated by gelatin and cellulosic fibrils, the different ratios between components will modify the shape of the emission band. Moreover, the blue emission of cellulose is highly dependent on crystallinity degree, and manual papers present large variation between manufacturers, due to rags type and quality [68], and due to technology [69,70].

Surprisingly, the treated samples had a quenched fluorescence after ZnO nanoparticles entered into the cellulosic net. Despite the high fluorescence of ZnO, the treated paper samples presented a few lower intensity emission bands. The NBE emission intensity was 3–6 times lower when compared with ZnO powder, while the DLE decreased by 25–50% only. ZnO nanoparticles can interact with -OH groups from cellulose fibrils. but also from collagen degradation products (gelatin) [71,72]. The same interactions between ZnO nanoparticles and cellulose fibrils that enhance the UV-band absorption intensity (in UV-Vis spectra, Figure 11) are responsible for the decrease of the fluorescence emission.

### 3.9. Antimicrobial Assay for Paper Samples before and after Treatment with ZnO

Our results revealed that ZnO treatment of old book samples, with visible microfungi colonization and degradation, completely removes the colonizing species (both mycelia and spores). In paper-treated samples, the growth of microfungi was absent in PDA after 7 days of incubation in standard conditions. As expected, in untreated paper samples with mold colonization and biodegradation aspect, we have isolated multiple microfungal species, such as *Aspergillus niger*, *Trichoderma* sp. and *Penicillium* sp., which are known human opportunistic pathogens (Figure 14).

*Aspergillus sp*. and *Penicillium* sp. are the most important mold geniuses with great implications in human and animal pathology, causing numerous skin and respiratory infections [73].

In Figure 14, it can be observed a combined growth of various filamentous fungi in the untreated old book paper sample A (Figure 14a,d) and a unique microfungal species in the plates where the untreated old paper specimen B (Figure 14b,e) was seeded. In the PDA plates inoculated with ZnO NPs treated old book paper sample A_ZnO (Figure 14c,f), no fungal development has been noticed. Antifungal activity of ZnO is reported in fewer papers if compared with antibacterial activity. Nevertheless, there are reports of activity against *Aspergillus niger* [56,74], *Trichoderma* sp. [75,76] and *Penicillium* sp. [77,78].

The paper sample A, belonging from a book, was more contaminated than unused manual paper sample B. The reason for this mix of fungal strains, *Aspergillus niger*, *Trichoderma* sp. and *Penicillium* sp., can be related to the fact that the book was handled by users, while the blanc paper was only stored in library.

## 4. Conclusions

ZnO nanoparticles, obtained by the solvothermal method, are polyhedral shaped, with a ~20 nm diameter and high crystallinity. The investigation of photocatalytic activity against methylene blue indicated that ZnO nanopowder has the capacity to degrade both the MB dye monomer and the dimer, under visible light irradiation. This high photocatalytic activity imparts a good antimicrobial activity. Indeed, the ZnO nanopowder exhibited good antibacterial activity against both Gram-positive and Gram-negative strains. The best results were observed against *Enterococcus Faecalis* (ATCC 44479) and *Escherichia coli* (ATCC 25922), while the weakest activity was recorded against *Pseudomonas aeruginosa* (ATCC 27853). This synthesis route allows the obtaining of pure ZnO at nano scale dimensions, exhibiting antibacterial and antifungal activities dependent of their particle size, surface chemistry, and defects generated as seen also in photocatalytic processes.

Treatment of paper samples from XVIII-XIX century with ZnO nanoparticles indicated that the ZnO nanopowder can also clean and protect the cellulosic material from fungal infections with *Aspergillus niger*, *Trichoderma* sp. and *Penicillium* sp., strains, present onto untreated paper samples. The optical properties of treated paper were enhanced, and ZnO can be used as UV-barrier to protect the cellulose from the harmful radiation.

## Figures and Tables

**Figure 1 materials-13-05452-f001:**
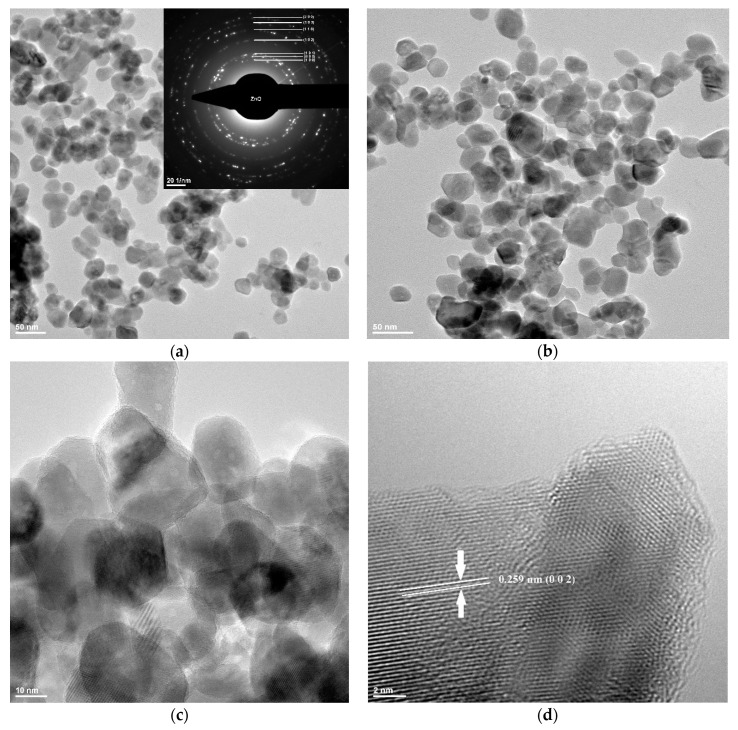
TEM (**a**–**c**) and high-resolution transmission electron microscopy (HRTEM) (**d**) images for ZnO nanoparticles.

**Figure 2 materials-13-05452-f002:**
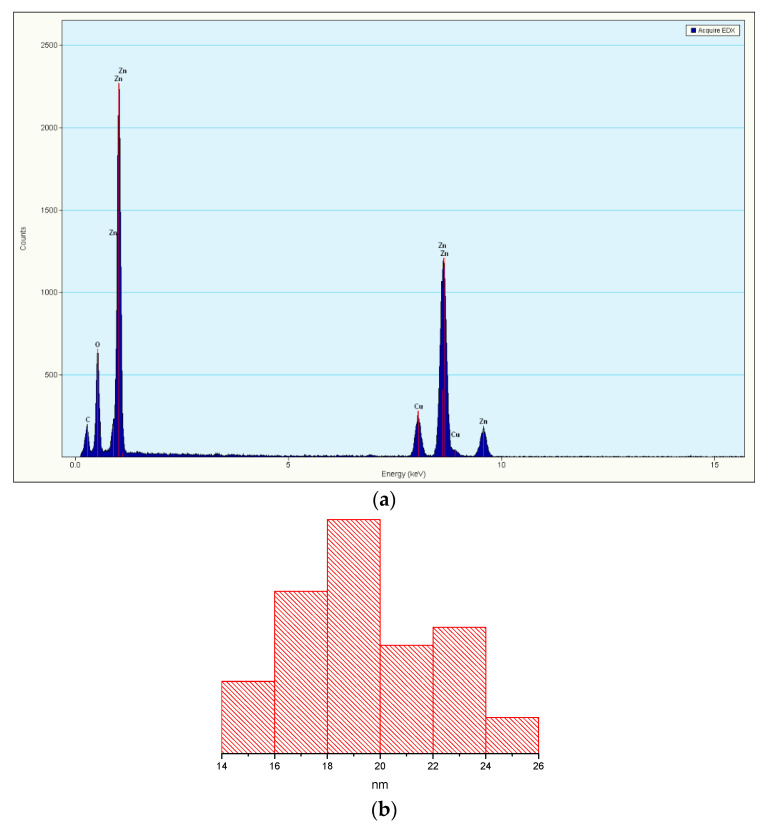
(**a**) EDX spectrum with Zn Lα and Kα lines (Cu lines are due to the copper grid used for placing the sample); (**b**) particle size distribution histogram.

**Figure 3 materials-13-05452-f003:**
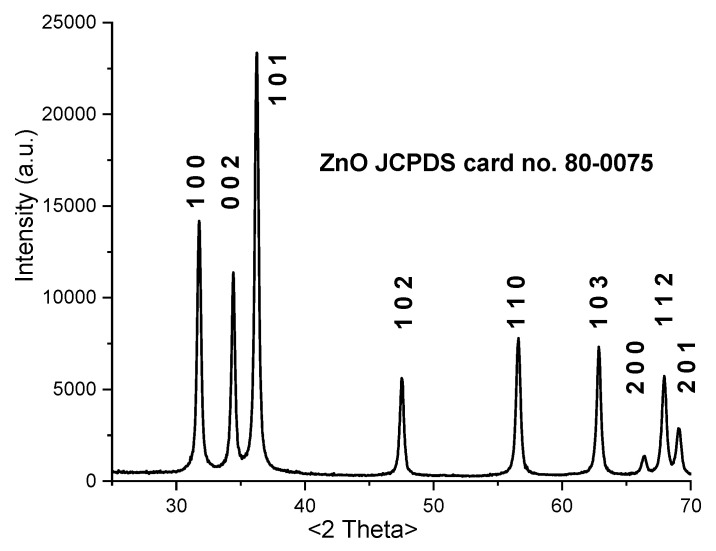
X-ray diffraction pattern for the ZnO nanoparticles.

**Figure 4 materials-13-05452-f004:**
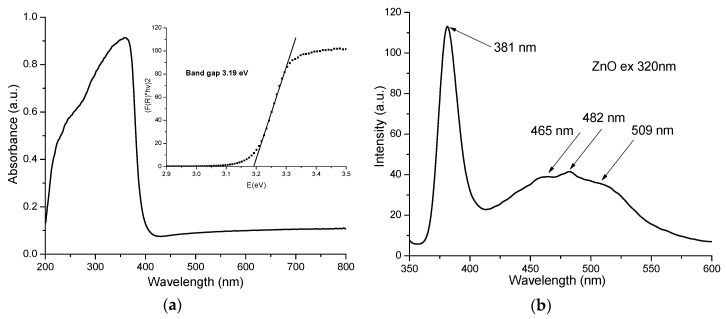
UV-Vis spectrum of ZnO; (**a**) with band-gap determination (inset); fluorescence spectrum of ZnO for excitation wavelength 320 nm (**b**).

**Figure 5 materials-13-05452-f005:**
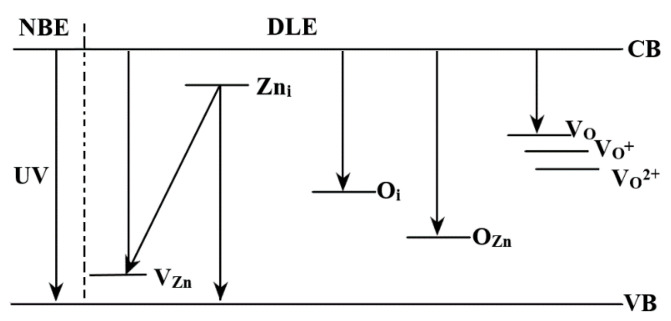
Defect’s levels in ZnO nanoparticles; the UV emission is referred to as Near Edge Band (NBE), while the visible emission is caused by Deep Level Emission (DLE).

**Figure 6 materials-13-05452-f006:**
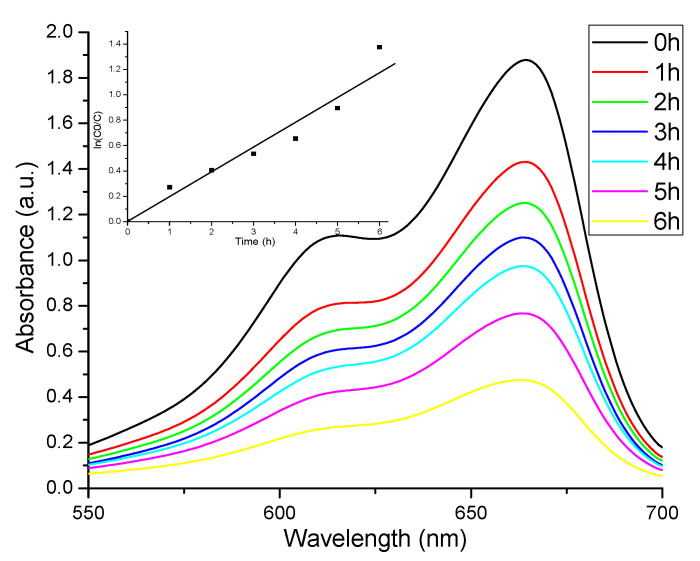
Photocatalytic activity against methylene blue (MB); the pseudo-first-order rate constant *k*_app_ (h^−1^) was calculated from the slope of ln(C0/Ct) versus irradiation time t.

**Figure 7 materials-13-05452-f007:**
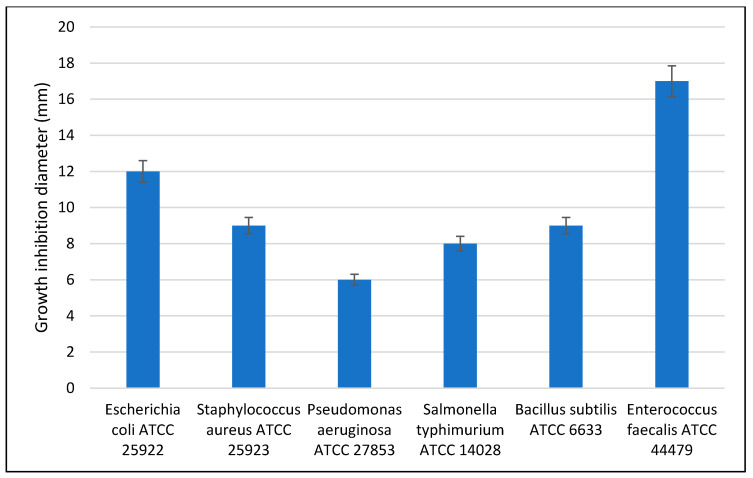
The antimicrobial activity of ZnO nanoparticles.

**Figure 8 materials-13-05452-f008:**
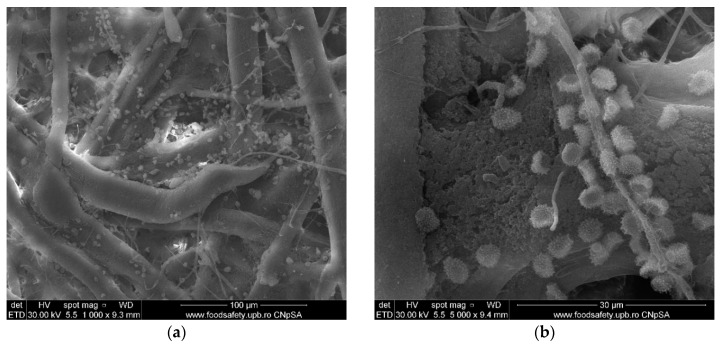
Different microorganisms found in paper sample A (**a**,**b**) and sample B (**c**,**d**).

**Figure 9 materials-13-05452-f009:**
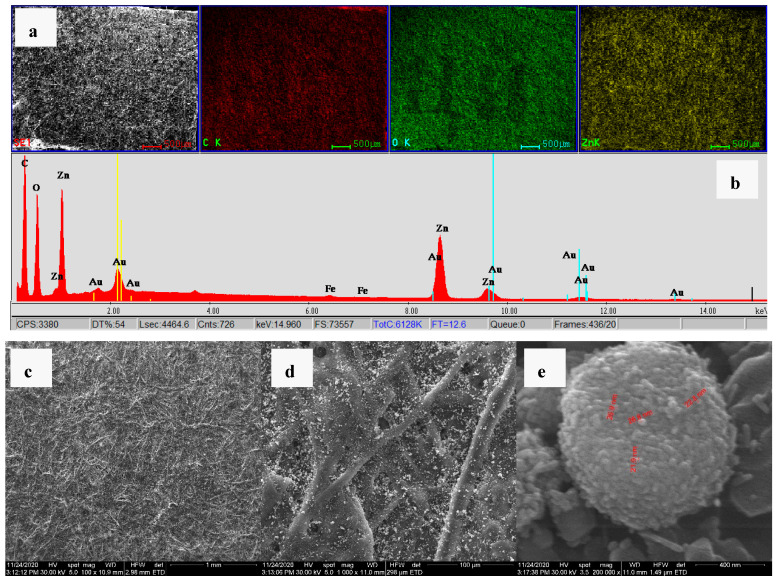
SEM and EDS elemental mapping for ZnO treated paper. SEM image (**a**) with elemental distribution: carbon–red; oxygen–green and zinc–yellow at 60×. Energy dispersive X-ray spectrum (**b**). EDX spectrum of the treated sample, with lines from C, O and Zn as elements composing the sample (Au lines are due to conducting thin film deposition), (**c**) SEM image at 100× magnification, (**d**) SEM image at 1000× magnification with clear ZnO agglomerates inside the fibrillar structure of paper, (**e**) SEM image at 200k× magnification with individual nanoparticles identification.

**Figure 10 materials-13-05452-f010:**
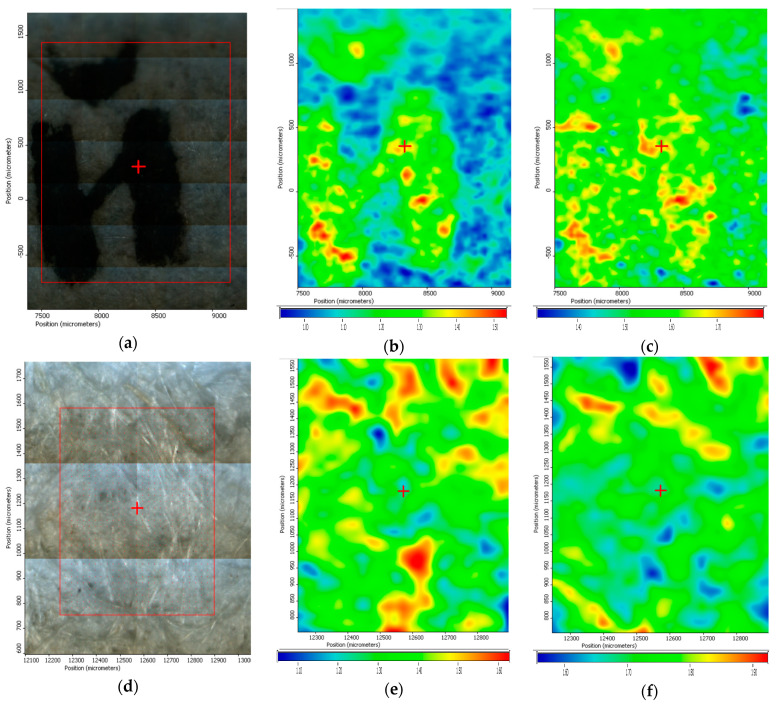
FTIR microscopy on sample A_ZnO (**a**–**c**) and sample B_ZnO (**d**–**f**): video image of paper (**a**,**d**); FTIR map at 1643 cm^−1^ (**b**,**e**); FTIR map at 430 cm^−1^ (**c**,**f**).

**Figure 11 materials-13-05452-f011:**
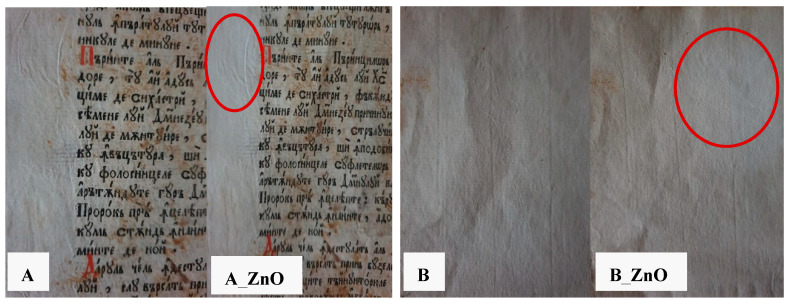
The original paper A and B and treated paper A_ZnO and B_ZnO.

**Figure 12 materials-13-05452-f012:**
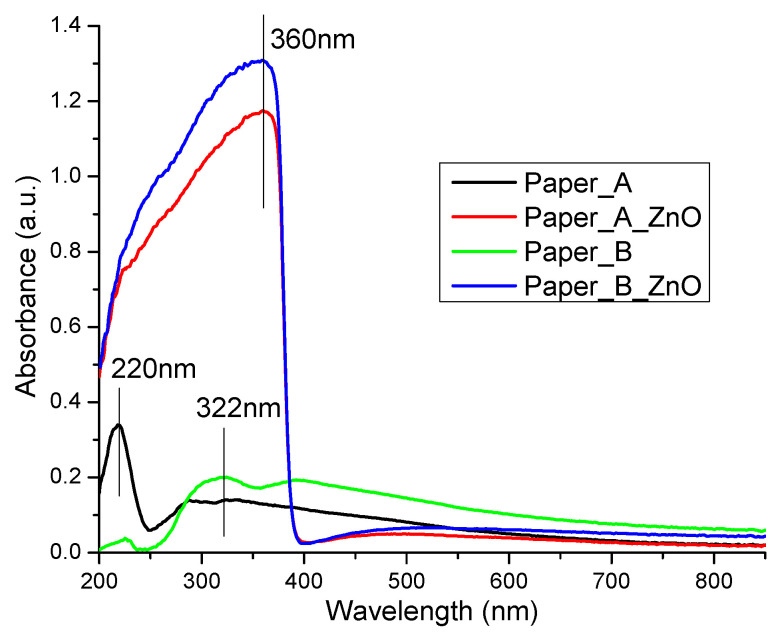
UV-Vis spectra for paper samples before and after ZnO treatment.

**Figure 13 materials-13-05452-f013:**
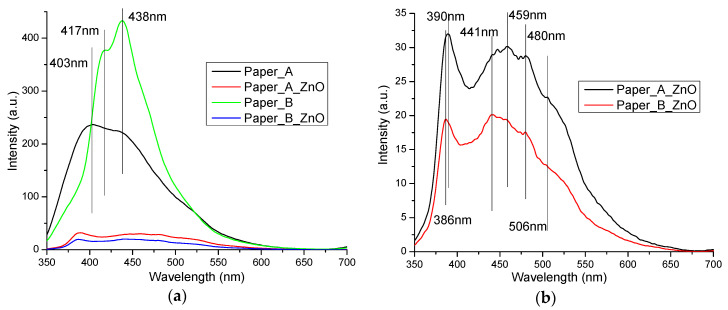
Photoluminescence spectra for paper samples before and after ZnO treatment (**a**); details of fluorescence spectra for ZnO treated paper (**b**).

**Figure 14 materials-13-05452-f014:**
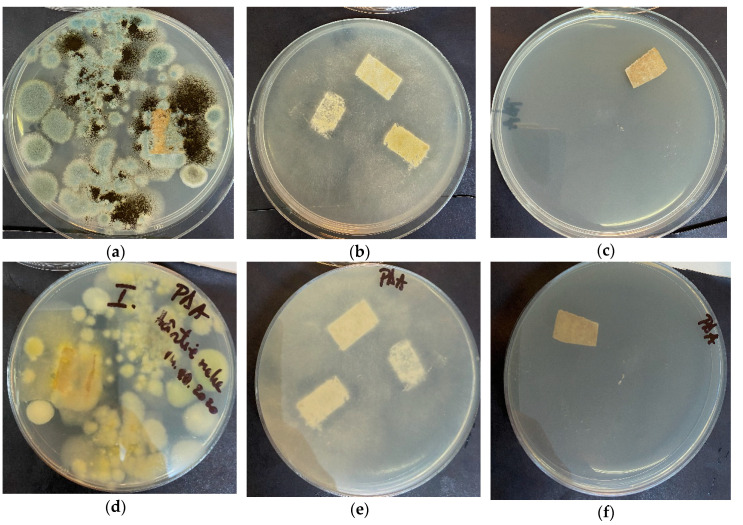
Aspect of mold cultures obtained in potato dextrose agar (PDA) after three-point inoculation using untreated and ZnO NPs treated old book specimens. Sample A (**a**,**d**) paper specimen with microfungi colonization and biodegradation aspect (a—surface; d—reverse); sample B (**b**,**e**) paper specimen with microfungi colonization and biodegradation aspect (b—surface; e—reverse); sample A_ZnO (**c**,**f**) treated paper specimen (c—surface; f—reverse).

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
