# Peer review of "Facile Use of ZnO Nanopowders to Protect Old Manual Paper Documents"

_materials, 2020, doi:10.3390/ma13235452_

Round 1

Reviewer 1 Report

Facile use of ZnO nanopowders to protect old manual paper documents is important study. However, there is commentes should be improved befor accepted:

  • the section of "deep level emission (DLE)" in page 6 and 7 and figure 5 should be removed from manuscript; it is not be possible to form oxygen vacancy in this method and should be improved by XPS and EPR; if you have it; or remove this part from manuscript and this hypothesis or you should do XPS and EPR.
  • all figures should be drawn and labeled as (a and b) not as (right and left) as example in figure 4.
  • The authors calculated the crystallite size using Debye-Scherrer's formula without considering the broadening of the diffractometer itself and microstrain in the materials. The crystallite size should be re-calculated after correcting the broadening of the diffractometer using standard materials. 
  • improve the introduction part with more details about significance of ZnO; as presented in some literature doi.org/10.1515/msp-2018-0045 and doi.org/10.1016/j.porgcoat.2016.12.026.
  • change all concentration to be by mol "as in page 3 line 102".
  • clarify the radiation source range in UV and visible or UV only, or detect the range of wavelength range of lamp used in MB degradation.
  • compare the activity of catalyst with other literature.

Author Response

Point 1: All figures should be drawn and labeled as (a and b) not as (right and left) as example in figure 4. Change all concentration to be by mol "as in page 3 line 102".

Response 1: We are thankful for pointing out this weakness. Corrections indicated were done. The figures 2, 4, 8, 10, 12, 13 were labeled with letters and the caption was changed accordingly. The concentration was given in mol but also in mg/L in order to facilitate the comparation with other results from literature as requested in point 5.

Point 2: The authors calculated the crystallite size using Debye-Scherrer's formula without considering the broadening of the diffractometer itself and microstrain in the materials. The crystallite size should be re-calculated after correcting the broadening of the diffractometer using standard materials. 

Response 2: By taking the advice of the reviewer we made the pattern again using PANalytical Empyrean equipment (from Malvern PANalytical, Bruno, Nederland) using a Bragg–Brentano geometry, equipped with a Cu anode (λCuKα = 1.54184 Å) X-ray tube and hybrid monochromator (Ge220). The X-ray diffractogram (XRD) was acquired in the 2θ range 10–75. After that the obtained data was fitted by Rietveld refinement in order to determinate the crystallite size and microstrain. The obtained values were 26.75±3.038 nm for crystallites size and 0.164±0.0509% for microstrain. We use as standard materials the ASTM file of pure wurtzite ZnO. We do not consider that there is any need to use standard materials because we did not make quantification of phases (as only one phase was identified). The new diffractogram and values were inserted in section 3.2.

Point 3: Improve the introduction part with more details about significance of ZnO; as presented in some literature doi.org/10.1515/msp-2018-0045 and doi.org/10.1016/j.porgcoat.2016.12.026.

Response 3: We are thankful to the esteem reviewer for giving us the opportunity to enrich the manuscript by providing more information on ZnO. The indicated papers and other relevant ones were used to describe the domains and applications for ZnO. (rows 61-64 and further in section 1).

Point 4: Clarify the radiation source range in UV and visible or UV only, or detect the range of wavelength range of lamp used in MB degradation.

Response 4: The source used is a commercially available fluorescent lamp, LOHUIS®, of 160W / 2900 lm, with color temperature of 3200K and color rendering index > 60. The information was added to 2.2 section.

Point 5: Compare the activity of catalyst with other literature.

Response 5: We are thankful for this suggestion. Comparison with results reported in literature for MB solutions with 10-15 mg/L concentrations and photocatalyst amount 50-75 mg were added in section 3.4.

Point 6: The section of "deep level emission (DLE)" in page 6 and 7 and figure 5 should be removed from manuscript; it is not be possible to form oxygen vacancy in this method and should be improved by XPS and EPR; if you have it; or remove this part from manuscript and this hypothesis or you should do XPS and EPR.

Response 6: The literature presents enough examples of ZnO obtained by solvothermal methods that exhibit blue-green luminescence, which also in literature is related to the various defects, oxygen vacancies, zinc vacancies, zinc interstitial, oxygen interstitial etc.

Please see: Journal of Sol-Gel Science and Technology 29, 71–79, 2004 - Non-Basic Solution Routes to Prepare ZnO Nanoparticles;

J Nanopart Res - Evolution of the zinc compound nanostructures in zinc acetate single-source solution; doi 10.1007/s11051-011-0504-y

New J. Chem., 2019,43, 17980-17990 - Understanding the role of alcohols in the growth behaviour of ZnO nanostructures prepared by solution based synthesis and their application in solar cells; https://doi.org/10.1039/C9NJ03212F

International Journal of Pharmaceutics 463 (2014) 161– 169 - Synthesis and characterization of a novel controlled release zinc oxide/gentamicin–chitosan composite with potential applications in wounds care - doi.org/10.1016/j.ijpharm.2013.11.035

For XPS results on ZnO sample prepared in methanol please see: Digest Journal of Nanomaterials and Biostructures 2011, 6(3) 1393-1401 "Synthesis and characterization of ZnO nanopowder by non-basic route"

Also on this topic with conclusion from the article:

doi.org/10.1016/j.jscs.2017.09.004 (annealing is removing the defects from solvothermal synthesis); doi.org/10.3390/ma3042643 (growth at temperatures as low as 50 °C has been demonstrated and shown to yield ZnO nanorods with excellent visible luminescence);

Reviewer 2 Report

The manuscript by Motelica et al reports on the use of ZnO nanoparticle as antimicrobial agents for old paper .  

The subject of the article can be of interest to Materials readers and the paper is sufficiently clear and well-written, even if the manuscript can benefit from a careful revision.

The following requests should be properly addressed by the authors before considering the manuscript for publication:

  1. The solvent in which particles are dispersed for application purposes should be indicated because it can have a fundamental role in the application of this material to paper. In other terms, are the particles dispersed in water? If so, what is the pH of the suspension? If water is not the dispersing media, are particles dispersed in ethanol (the solvent used for the synthesis)? In any case, authors should add this piece of information. It must be noted that alkaline solutions or ethanol might have themselves an antimicrobial effect and this should be evaluated with additional tests.

  1. Another information that is currently missing is the amount of treatment used on paper samples, in terms of g/cm2 or mL/cm2. In the latter case, concentration should be also added.

  1. In Figure 2 (right), histograms showing the size distribution of particles are reported. Some information about how these data were obtained must be included, especially because in the TEM pictures (Figure 1a and 1b) particles of about 35-50 µm are also clearly shown.

  1. About the photocatalytic activity, I do not think that the photodegradation reaction exhibit a first order kinetic. What is the R2 of the linear regression reported in Figure 6? I think that authors should considered higher order kinetics.

  1. The FTIR mapping discussion could be expanded to help the readers understand the experimental data. For instance, a comparison between FTIR spectra of ink, untreated paper and treated samples should be included, to clearly show why authors decided to collect the map at 430 cm-1 and 1643 cm-1, which, I suppose, is related the presence of some functional groups that are characteristic of specific components. This must be indicated and discussed. In addition to that, a color scale bar must be included, and some comments must be added to clarify how maps are interpreted.

  1. It would be very interesting to have some information about the color changes in samples due to the application of ZnO particles. Authors can use the VIS reflectance spectra they already collected to extract the colorimetric coordinates and calculate the DeltaE values.

Additional comments:

  1. Image quality of Figures 9 and 13 is poor.

  1. Lines 130-132 are misplaced, because they anticipate some of the results. Therefore, I suggest moving them at the end of the discussion or in the Conclusion section.

  1. Figure 2, EDX spectrum. I suppose that a copper grid was used for TEM acquisition, due to the fact that copper signals are present in the spectrum. If so, add this piece of information in section 2.2 (preparation of samples) or add a comment in the figure caption.

Author Response

Point 1: The solvent in which particles are dispersed for application purposes should be indicated because it can have a fundamental role in the application of this material to paper. In other terms, are the particles dispersed in water? If so, what is the pH of the suspension? If water is not the dispersing media, are particles dispersed in ethanol (the solvent used for the synthesis)? In any case, authors should add this piece of information. It must be noted that alkaline solutions or ethanol might have themselves an antimicrobial effect and this should be evaluated with additional tests.

Response 1: We are very grateful for pointing out this omission. The ZnO was sprinkled on the paper sheet and evenly distributed after with a soft brush in linear moves. The excess powder was discarded by gentle tapping. The information was added in 2.1 section.

Point 2: Another information that is currently missing is the amount of treatment used on paper samples, in terms of g/cm2 or mL/cm2. In the latter case, concentration should be also added.

Response 2: We thank to the esteem reviewer for pointing out the missing information and helping us to improve the manuscript. The amount of ZnO retained on the paper surface is 0.23 ± 0.011 mg/cm2 on each side. Information was added in rows 163-165.

Point 3: In Figure 2 (right), histograms showing the size distribution of particles are reported. Some information about how these data were obtained must be included, especially because in the TEM pictures (Figure 1a and 1b) particles of about 35-50 µm are also clearly shown.

Response 3: The particle size distribution of the ZnO was obtained by measuring about 500 np’s. The images itself shows larger particles indeed, but, if check in detail, that bigger particles are in fact 2 or 3 nanoparticles overlapping. Only the particles that are clearly defined by the edges were taken into account for particle size distribution and the most extreme ranges were discarded from statistical reasons. But as the esteem reviewer pointed out, there might be bigger particles in this nanopowder, without implications on the obtained results or the described application. We added this information in section 3.1. rows 180-182, and at rows 206-207.

Point 4: About the photocatalytic activity, I do not think that the photodegradation reaction exhibit a first order kinetic. What is the R2 of the linear regression reported in Figure 6? I think that authors should considered higher order kinetics.

Response 4: We have added literature information in section 3.4. about results obtained for same system (ZnO/MB). We are thankful for this observation. The value calculated with Origin 8.6. software is R-square = 0.940. The information was added in section 3.4 together with comparison with literature data.

Point 5: The FTIR mapping discussion could be expanded to help the readers understand the experimental data. For instance, a comparison between FTIR spectra of ink, untreated paper and treated samples should be included, to clearly show why authors decided to collect the map at 430 cm-1 and 1643 cm-1, which, I suppose, is related the presence of some functional groups that are characteristic of specific components. This must be indicated and discussed. In addition to that, a color scale bar must be included, and some comments must be added to clarify how maps are interpreted.

Response 5: Our first concern was that ZnO nanopowder could impact the clarity of the letters printed on the old paper, and lead to information loss. Therefore, the FTIR maps were added as a proof for uniform distribution of ZnO nanoparticles, and to indicate that even after treating the paper the letters were unaffected, being clearly visible. Information was added to the 3.7. section. Color scale bars were added to each picture.

Point 6: It would be very interesting to have some information about the color changes in samples due to the application of ZnO particles. Authors can use the VIS reflectance spectra they already collected to extract the colorimetric coordinates and calculate the DeltaE values.

Response 6: Unfortunately, the UV-Vis spectrophotometer, Jasco V 560, does not have the possibility to transform the spectral data collected in CIE l*a*b so we can calculate ΔE values. By using some online available calculators, the ΔE value obtained is ~13 for A/A_ZnO and ~15 for B/B_ZnO. As we did not use a scientific software for this transformation, we think is better not to report these values in the article.

Point 7: Image quality of Figures 9 and 13 is poor.

Response 7: Unfortunately, the images are taken with the best camera we have at our disposal. Somewhere in the MDPI electronic submission system the picture resolution was decreased, as original word file had 38 Mb and now the word received for revision has only 22 Mb. We have put back the original pictures into the reviewed manuscript, and also the original EDX spectra at figure 2a.

Point 8: Lines 130-132 are misplaced, because they anticipate some of the results. Therefore, I suggest moving them at the end of the discussion or in the Conclusion section.

Response 8: The phrase was moved at Conclusion section.

Point 9: Figure 2, EDX spectrum. I suppose that a copper grid was used for TEM acquisition, due to the fact that copper signals are present in the spectrum. If so, add this piece of information in section 2.2 (preparation of samples) or add a comment in the figure caption.

Response 9: We are thankful to the reviewer for point out the missing information. Indeed, a copper grid was used for placing ZnO sample into the TEM. Information was added in figure 2a caption.

Reviewer 3 Report

The paper by Motelica et al. is devoted to the potential application of ZnO nanopowder as an antibacterial agent mixed with the fiber tissue of aging paper, in that way protecting old printed books from biological contamination.

The idea is well and clearly presented, the manuscript is written in a good language. The background, as concerning the proposed antibacterial influence of the oxide nanoparticles, is also put down clearly enough. Experimental procedures are well divided and presented. The figures are clear and the conclusions seem to basically confirm the hypotheses raised.

However, there remains only one issue which at the moment seems to be ignored (if the referee is not heavily mistaken now) in the present version of the manuscript. Namely, the authors have hardly addressed the potential hazards of ZnO particles to human health. This is kind of important, because these particles are meant to be used in environments embracing humans, i.e. the actual users of those old manual paper documents.

There seems to exist a considerable body of literature, dealing with examination of ZnO nanoparticles as potentially dangerous species. There can be works found, claiming toxicity of ZnO nanoparticles towards human lung cells [Sahu et al. http://dx.doi.org/10.1155/2013/316075],  neurotoxic potential of ZnO particles [Sruthi et al. https://doi.org/10.1016/j.mtchem.2018.09.008 ], cytotoxicity of ZnO nanoparticles in general [Vandebriel and Jong, http://dx.doi.org/10.2147/NSA.S23932 ], hepatotoxic effect (still on animals) [El Shemy et al. http://dx.doi.org/10.11648/j.ijbbmb.20170201.11 ], cytotoxicity towards human kidneys [Reshma and Mohanan http://dx.doi.org/10.1016/j.colsurfb.2017.05.069 ],

Of course, there are also some papers, which attempt to prove the safety of ZnO nanoparticles. For instance, in a study by Mohammed et al., http://dx.doi.org/10.1016/j.jid.2018.08.024 , human volunteers were involved in the experiments, and it was concluded that the ZnO particles are at least dermatologically not harmful. This actually seems to be quite important also from the point of view of the present manuscript. A study by Alaraby et al. [http://dx.doi.org/10.1016/j.jhazmat.2015.04.053 ] summarized that ZnO particles did not induce genotoxicity in vinegar fly, although altered genes were detected. Another paper by McClements and Xiao [http://dx.doi.org/10.1038/s41538-017-0005-1 ] seems to explain, that – since there are a lot of particles, including those of ZnO, present in the food industry anyway, they are not supposed to be harmful to humans. Still, the author of the latter reference also admit, that future studies should be conducted in this regard.

In general, it seems that the amount of papers claiming toxicity of ZnO nanoparticles somewhat exceeds the amount of papers believing in their safety.

In summary, it is strongly recommended (actually obligatory), that the authors of the manuscript tell an arbitrary reader about how convinced they are that introduction of antibacterial ZnO particles into objects to be placed in the same room with human beings will not cause an additional threat to the humans health. It is also recommended that the authors enclose a separate paragraph in the Introduction section, addressing that matter. Please note, that this would not reduce the scientific credibility of this particular manuscript, but just increase its integrity and reader’s ability to consider all the positive and negative aspects! The paper is otherwise indeed interesting and worth publishing.

Author Response

Response: We thank the reviewer for useful suggestions. We included a paragraph, in section 1, addressing the risk of human exposure by inhalation or dermal contact of ZnO NPs and we cited some of the references pointed out in the comment. We added them, together with other references found relevant by us, to the list of references.

“Generally, ZnO nanoparticles (NPs) or bulk ZnO particulates, when used as antibacterial agents, are considered bio-safe within certain limits which depend on the availability or concentration of zinc ions, but rise safety issues at higher concentrations [18], especially in nano form when ingested [19]. ZnO NPs have widespread use in cosmetics [20], various paints and coatings [9], active food packaging [21] and numerous biomedical applications [22], and, therefore, hazards from exposure via inhalation [23] and dermal exposure to them are of great concern and numerous studies have been carried out assessing their cytotoxicity when ingested or applied onto human skin [24-26]. The results are still controversial because of the information scarcity on human exposure assessment in realistic use [27], since the majority of studies were focused on the first two steps of risk assessment process: the intrinsic toxicity and dose-response relationship of ZnO NPs [28, 29]. Although, ZnO NPs exhibited in vitro experiments oxidative stress-induced apoptosis in human lung epithelial cells (L-132) [23] or neurotoxic potential on neuronal cells [24], the cytotoxicity of ZnO NPs in these studies is evaluated at concentrations that are likely higher than would be expected for short-term exposure (topical or inhalation) [30]. There is widely accepted that ZnO NPs do not penetrate healthy human skin, even they do affect skin cells in vitro [25, 27]. Furthermore, all in vivo and most in vitro genotoxicity tests were negative for ZnO NPs [28, 31, 32].”

Reviewer 4 Report

All formulas and diagrams should be moved to the part concerning the methodology. Detailed information on the deposition of ZnO particles to paper samples should be provided. The resolution of figure 2 (EDX) should be improved. It is desirable to identify microorganisms in Fig. 8. The figure is interesting, but far from the research topic (fungus, bacteria). Figure 9 and figure 10-the captions are misleading. It should be clearly indicated where the initial sample (or samples) is, and where the sample is after processing with nanoparticles. It is necessary to use a letter designation: a. b. c. d. Difficult to read "top, bottom, middle..."

Author Response

Point 1: Detailed information on the deposition of ZnO particles to paper samples should be provided.

Response 1: We are very grateful for pointing out this omission. The ZnO was sprinkled on the paper sheet and evenly distributed after with a soft brush in linear moves. The excess powder was discarded by gentle tapping. The information was added in 2.1 section (rows 104-106). The amount of ZnO retained on the paper surface is 0.23 ± 0.011 mg/cm2 on each side (rows 163-165).

Point 2: The resolution of figure 2 (EDX) should be improved.

Response 2: We have replaced the figure 2 with the original one which has a better resolution. Somewhere in the MDPI electronic submission system the picture resolution was decreased, as original file had 38 Mb and now the word file has only 22 Mb.

Point 3: It is desirable to identify microorganisms in Fig. 8. The figure is interesting, but far from the research topic (fungus, bacteria).

Response 3: Aspergillus niger, Trichoderma sp. and Penicillium sp. were identified primarily according to the culture and the microscopic images were additionally recorded to show the high level of contamination with these fungal strains on the old book paper samples. However, based on microscopic characteristics we can identify the presence of Aspergillus sp. and we completed added the information at section 3.6 row 308.

Point 4: Figure 9 and figure 10-the captions are misleading. It should be clearly indicated where the initial sample (or samples) is, and where the sample is after processing with nanoparticles.

Response 4: We are thankful for pointing out this misunderstanding possibility. We moved figure 9 to the UV-Vis/PL section (3.8) to which is related, as the UV-Vis and PL spectra were measured on indicated area. Those pictures were made to better visualize the impact of ZnO application, and therefore initially the treatment was only on limited area. In UV-Vis and PL spectroscopy we needed parts of paper with no ink, in order to eliminate the possible interferences from ink (the area measured in one run is about 1 cm2). In FTIR microscopy the maps are recorded point by point and we know exactly what data is coming from ink zone and what data is recorded from paper around letter. Our first concern was that ZnO nanopowder could impact the clarity of the letters printed on the old paper, and lead to information loss. The FTIR maps were recorded after we treated also the printed area, and we wish to demonstrate that the ink is clearly visible after ZnO treatment. Therefore, the FTIR maps were added as a proof for uniform distribution of ZnO nanoparticles, and to demonstrate that even after treating the paper the letters were unaffected, being clearly visible. Information was added to the 3.7. section.

Point 5: It is necessary to use a letter designation: a. b. c. d. Difficult to read "top, bottom, middle..."

Response 5: We are thankful for pointing out this weakness. Corrections indicated were done. The figures 2, 4, 8, 10, 12, 13 were labeled with letters and the caption was changed accordingly.

Point 6: All formulas and diagrams should be moved to the part concerning the methodology.

Response 6: The Scherrer equation was deleted as the XRD data was collected again and fitted by Rietveld refinement. The Kubelka-Munk equation, is part of how obtained data are further processed to obtain the band gap presented in figure 4a (inset). We think that the esteem reviewer is referring to the figure 5 as diagram. The figure 5 is depicting in detail the transitions that generate the fluorescence spectra and therefore we feel that it should be near the text where those transitions are explained.

Reviewer 5 Report

The manuscript ‘Facile use of ZnO nanopowders to protect old manual paper documents’ is a very interesting paper and has important practical aspects, but I believe that manuscript in present form, should not be published in the Materials. It is necessary to preparing the revised version of this articles and I have a few comments that the authors should consider in preparing the revised version.

Results and discussion section was divided into two parts. The first parts concerns to characterisation of ZnO nanopowders using different techniques, such as TEM, XRD, UV-VIS and examination of antimicrobial and photocatalytic activities. The novelty of these results is moderate, because such studies have already been repeatedly published by various authors. The second part of this section concerns the testing of paper modified with ZnO nanopowders and contains novelty elements. It’s imperative that the authors complete results in this part of Results and discussion section. Authors should be explain:

  1. How method and parameters are used to modified of papers?
  2. Why another techniques are used to characterisation of ZnO nanopowders and ZnO-modified paper?
  3. How ZnO nanoparticles are bond to the cellulose surface?
  4. Whether ZnO released from cellulose surface will negatively affect paper users?

Moreover, Authors should add of SEM images of the ZnO-modified paper and compare with the SEM results of the unmodified paper.

Author Response

Point 1: How method and parameters are used to modified of papers?

Response 1: We are very grateful for pointing out this omission. The ZnO was sprinkled on the paper sheet and evenly distributed after with a soft brush in linear moves. The excess powder was discarded by gentle tapping. The information was added in 2.1 section (rows 104-106). The amount of ZnO retained on the paper surface is 0.23 ± 0.011 mg/cm2 on each side (rows 166-169).

Point 2: Why another techniques are used to characterisation of ZnO nanopowders and ZnO-modified paper?

Response 2: We used same techniques (UV-Vis and fluorescence spectroscopy, and electronic microscopy) for both ZnO nanopowder and paper. As paper has a microstructure of fibers, we had no reason to go for TEM, SEM being more suited for its investigation. In the same time ZnO nanopowder is clearly in nano size zone and we wanted to have it better characterized than SEM will do. For ZnO we did investigate the photocatalytic activity, due to implication in antimicrobial activity mechanisms (ROS generation). For the ZnO nanoparticles there were more analyses done, but not presented in this article as they are not relevant for the scope. FTIR microscopy on a single phase nanopowder (ZnO) will give us only the information that there is a homogenic phase, as maps are recorded point by point, and everywhere will be only ZnO.

Our first concern was that ZnO nanopowder could impact the clarity of the letters printed on the old paper, and lead to information loss. The FTIR maps were recorded after we treated the printed area, and we wish to demonstrate that the ink is clearly visible after ZnO treatment. As in FTIR microscopy the maps are recorded point by point and we can visually know exactly what data is coming from ink zone and what data is recorded from paper around letter. Therefore, the FTIR maps were added as a proof for uniform distribution of ZnO nanoparticles, and to demonstrate that even after treating the paper the letters were unaffected, being clearly visible. Information was added to the 3.7. section.

Point 3: How ZnO nanoparticles are bond to the cellulose surface?

Response 3: ZnO NPs can be bonded by -OH groups of the cellulose. Information with relevant literature data was introduced at rows: 347-348. The fibrillar network also acts like a net, trapping the nanoparticles. We are thankful for pointing us this question.

Point 4: Whether ZnO released from cellulose surface will negatively affect paper users?

Response 4: We thank the reviewer for useful suggestions. We included a paragraph, in section 1, addressing the risk of human exposure by inhalation or dermal contact of ZnO NPs and we cited some of the references pointed out in the comment. We added them, together with other references found relevant by us, to the list of references.

“Generally, ZnO nanoparticles (NPs) or bulk ZnO particulates, when used as antibacterial agents, are considered bio-safe within certain limits which depend on the availability or concentration of zinc ions, but rise safety issues at higher concentrations [18], especially in nano form when ingested [19]. ZnO NPs have widespread use in cosmetics [20], various paints and coatings [9], active food packaging [21] and numerous biomedical applications [22], and, therefore, hazards from exposure via inhalation [23] and dermal exposure to them are of great concern and numerous studies have been carried out assessing their cytotoxicity when ingested or applied onto human skin [24-26]. The results are still controversial because of the information scarcity on human exposure assessment in realistic use [27], since the majority of studies were focused on the first two steps of risk assessment process: the intrinsic toxicity and dose-response relationship of ZnO NPs [28, 29]. Although, ZnO NPs exhibited in vitro experiments oxidative stress-induced apoptosis in human lung epithelial cells (L-132) [23] or neurotoxic potential on neuronal cells [24], the cytotoxicity of ZnO NPs in these studies is evaluated at concentrations that are likely higher than would be expected for short-term exposure (topical or inhalation) [30]. There is widely accepted that ZnO NPs do not penetrate healthy human skin, even they do affect skin cells in vitro [25, 27]. Furthermore, all in vivo and most in vitro genotoxicity tests were negative for ZnO NPs [28, 31, 32].”

Point 5: Moreover, Authors should add of SEM images of the ZnO-modified paper and compare with the SEM results of the unmodified paper.

Response 5: We can provide to the esteem reviewer SEM images of treated paper and elemental map but unfortunately, we cannot include them in the present article as they are part of a later unfinished study (please see attachment).

Round 2

Reviewer 1 Report

Accept in present form

Author Response

Thank you very much for reviewing our manuscript. We greatly appreciate your thoughtful comments and suggestions that helped improve the manuscript.

Reviewer 2 Report

I really appreciate that authors addressed most of my concerns regarding the manuscript. However, before being accepted for publication, I think that some additional clarifications/changes must be provided:

  1. Even if I appreciate the fact that additional information has been added, the FTIR mapping section is still confused. I will try to clarify my concern again.
    • In general, the same color scale must be selected for maps that are acquired on the same wavelength, such as those reported in Figure 9b and 9e, and in Figure 9c and 9f. Otherwise, a proper comparison is not feasible. This should be easily done using the FTIR software.
    • If I understand correctly, the maps at 430 cm-1 were acquired to check the distribution of ZnO particles over the treated surface, on an inked spot and on a non-inked spot. What is missing here, are the maps acquired using the same wavelength on non-treated samples, showing that no signals due to paper components are present at that wavelength. Those maps would be the blank or reference maps.
    • I honestly cannot still understand the rationale behind the maps acquired at 1643 cm-1. First of all, authors should try to clarify the attribution of the signal. Is it due to C=O stretching in amide I or to other paper/ink components? Why are they generally referring to a -N-H group (line 338) or to “various groups” (line 337)? Once the peak is attributed, they need to explain why in the inked part there is a strong increase in the signal, that we cannot see on non-inked paper. But more in general, in their reply, they stated that “our first concern was that ZnO nanopowder could impact the clarity of the letters printed on the old paper”. If they really want to address that concern, they need to provide a map acquired on a non-treated inked area, to check if there is a reduction in the intensity of the signals due to the application of ZnO nanopowder. If the same color scale is used, readers can clearly see if there is a change in the FTIR maps acquired on that wavelength.

  1. Authors need to carefully revise the English form of the paper, especially in the sections added during the revision.

  1. The way errors are reported must be corrected. For instance, at line 168, the corrected value to be indicate is 0.23 ± 0.01, and not 0.23 ± 0.011. In other terms, the number of meaningful digits of the error must agree with those used for the reported data.

Author Response

Thank you very much for reviewing our manuscript. We greatly appreciate your thoughtful comments and suggestions that helped improve the manuscript. We have carried out the suggested modifications and revised the manuscript accordingly. We hope you find the revised manuscript acceptable for publication. Thank you once again for your effort.

Point 1: Even if I appreciate the fact that additional information has been added, the FTIR mapping section is still confused. I will try to clarify my concern again. In general, the same color scale must be selected for maps that are acquired on the same wavelength, such as those reported in Figure 9b and 9e, and in Figure 9c and 9f. Otherwise, a proper comparison is not feasible. This should be easily done using the FTIR software. If I understand correctly, the maps at 430 cm-1 were acquired to check the distribution of ZnO particles over the treated surface, on an inked spot and on a non-inked spot. What is missing here, are the maps acquired using the same wavelength on non-treated samples, showing that no signals due to paper components are present at that wavelength. Those maps would be the blank or reference maps. I honestly cannot still understand the rationale behind the maps acquired at 1643 cm-1. First of all, authors should try to clarify the attribution of the signal. Is it due to C=O stretching in amide I or to other paper/ink components? Why are they generally referring to a -N-H group (line 338) or to “various groups” (line 337)? Once the peak is attributed, they need to explain why in the inked part there is a strong increase in the signal, that we cannot see on non-inked paper. But more in general, in their reply, they stated that “our first concern was that ZnO nanopowder could impact the clarity of the letters printed on the old paper”. If they really want to address that concern, they need to provide a map acquired on a non-treated inked area, to check if there is a reduction in the intensity of the signals due to the application of ZnO nanopowder. If the same color scale is used, readers can clearly see if there is a change in the FTIR maps acquired on that wavelength.

Response 1:

FTIR microscope is working a bit differently from the FTIR spectrometer as it presents the results under a map format, and we cannot normalize the results. For a specific wavenumber the transmittance can vary between sample’s points on the map (the microscope collects FTIR spectra point by point) due to different composition. The scale will be for that particular wavenumber adjusted from x to y values, which are the minimum and maximum values of transmittance obtained for the sample, for that wavenumber. Another sample will have a different interval in which transmittance will vary, hence the different numbers on the color bars, between samples. Even in same sample when we move to another wavenumber the min and max values of transmittance will change across the map, therefore we cannot keep the same scale on the color bars. The maps are interpreted for same sample and not between samples.

Indeed, after initial concern that ZnO might interfere with the clarity of the letters (which didn’t happen) we did the FITR map analysis to see if ZnO was retained on the printed area also, because we could clearly see the letters. Was the cheapest and fastest way for us to check. Since then, we have also confirmation of ZnO presence from SEM and EDS mapping analysis, which we have also added to the article now, in section 3.6.

We are grateful for pointing out the confusing interpretation. The 1643 cm-1 wavenumber is attributed to the amide band I, due to C=O bond stretching vibration (~80% of its intensity, with the rest from C-N groups and some small in-plane NH bending). Hemicellulose and lignin are not expected to be present in the old handmade paper, as it is made primarily from rags. Also bending of the absorbed water is located at this wavenumber. We have simplified the sentence to clearly point out the wavenumber attribution.

Point 2: Authors need to carefully revise the English form of the paper, especially in the sections added during the revision.

Response 2: We have thoroughly checked the English language and style. Mistakes were found and corrected, some paragraphs were rephrased to become clearer.

 Point 3: The way errors are reported must be corrected. For instance, at line 168, the corrected value to be indicate is 0.23 ± 0.01, and not 0.23 ± 0.011. In other terms, the number of meaningful digits of the error must agree with those used for the reported data.

Response 3: We are thankful for pointing out this weakness. The requested corrections were made across the manuscript where standard deviation was calculated.

Reviewer 4 Report

I liked the article. A comprehensive approach is visible. I even understand some of the words in the old book!

Author Response

(The authors gave the same response as above.)

Reviewer 5 Report

In present from, the manuscript can be accepted for publishing in Materials, because contains some new results and merits publication.

Author Response

(The authors gave the same response as above.)
